# New porous water ice metastable at atmospheric pressure obtained by emptying a hydrogen-filled ice

Leonardo del Rosso[1,2], Milva Celli[1] & Lorenzo Ulivi[1]

The properties of some forms of water ice reserve still intriguing surprises. Besides the several stable or metastable phases of pure ice, solid mixtures of water with gases are precursors of other ices, as in some cases they may be emptied, leaving a metastable hydrogen-bound water structure. We present here the first characterization of a new form of ice, obtained from the crystalline solid compound of water and molecular hydrogen called $C_0$-structure filled ice. By means of Raman spectroscopy, we measure the hydrogen release at different temperatures and succeed in rapidly removing all the hydrogen molecules, obtaining a new form of ice (ice XVII). Its structure is determined by means of neutron diffraction measurements. Of paramount interest is that the emptied crystal can adsorb again hydrogen and release it repeatedly, showing a temperature-dependent hysteresis.

[1] Consiglio Nazionale delle Ricerche, Istituto dei Sistemi Complessi, via Madonna del Piano 10, I-50019 Sesto Fiorentino, Italy. [2] Dipartimento di Fisica e Astronomia, Università degli Studi di Firenze, via Sansone 1, I-50019 Sesto Fiorentino, Italy. Correspondence and requests for materials should be addressed to L.U. (email: lorenzo.ulivi@isc.cnr.it).

Water molecules in the solid state may give rise to more than 15 different forms of ices, depending on temperature and pressure[1]. In addition, when water freezes in the presence of some other molecular substance that does not bind chemically to water, it may form peculiar crystal structures, known as clathrate hydrates, trapping guest molecules inside cages of different geometries[2]. Until recently, it has been believed that caged guest molecules are essential for the stability of the clathrate-hydrate crystals, so that the water skeleton would collapse without them. One experimental study has recently demonstrated, however, that at least one of these clathrate structures can be emptied of its guests, by letting neon atoms diffuse out of the solid, and persist in a metastable state if preserved at low temperature[3]. These and other similar low-density lattices of water molecules have raised recently large interest from a theoretical point of view, because they are believed to be the stable phase of solid water at negative pressure[4].

Water does form crystalline compounds with molecular hydrogen as well. Four ordered structures of this binary mixture are known. The first in order of increasing pressure, stable at $P > 100$ MPa and at 4–6 °C below 0, is a clathrate hydrate[5,6] having the so-called cubic sII structure, common to other clathrates with different molecular guests[7]. In this non-stoichiometric compound, hydrogen molecules are trapped in two types of cages with a total hydrogen molar fraction $X$ ($X = \mathrm{mol(H_2)/mol(H_2O)}$) up to ∼35%. The hydrogen molecules in the cages perform a peculiar quantum rattling and rotational motion, which has been studied efficiently with inelastic neutron scattering[8–12] and Raman scattering[13–16]. The latter technique is a very powerful and convenient one to identify the interactions of the molecules with the water environment and to measure the composition of the sample. Besides sII clathrates, the existence of two other stable phases for the $H_2$–$H_2O$ solid mixture at higher pressure ($P \gtrsim 700$ MPa) is known since 1993 (ref. 17). These two structures, indicated with $C_1$ and $C_2$, do not possess the typical cage structure of clathrates and are usually and more properly named filled ices. Recently, only two experimental studies[18,19] have investigated the phase diagram of the $H_2$–$H_2O$ compounds at intermediate pressures and have demonstrated the presence of a further stable phase, named $C_0$, at temperatures 100–270 K and pressures 360–700 MPa, which is intermediate between the stability region of the sII clathrate and $C_1$ phase (see Fig. 1). Up to now, a definitive consensus on the structure of the $C_0$ phase has not been reached. The structural model (later referred to as $C_0$-I) originally proposed in ref. 18 after ex-situ X-ray diffraction measurements at room pressure and 80 K, assumes space group $P3_112$ (or $P3_121$) and the presence of water molecules in sites $3a_2$ with 0.5 occupancy. These water molecules would not present correct hydrogen bonds with the other molecules of the lattice. Strobel et al.[19] suggest two possible structures (indicated as α-quartz and sT′) after the analysis of X-ray diffraction collected from one sample pressurized in a diamond anvil cell. These models were later examined theoretically by Smirnov and Stegailov[20], who considered also a variant of the $C_0$-I structure (indicated by $C_0$-II), and suggested $C_0$-II and sT′ both as viable candidates. In fact, the α-quartz structure, with the proposed parameters[20], gives rise to atypical, very short, O–O distances. After this, Oganov and colleagues[21] used density functional theory-based structure searching to predict several phases of the $H_2$–$H_2O$ mixture, but these searching methods are restricted in the number of atoms; thus, any large clathrate-sized unit cells can easily be missed. In this work we study, by means of Raman spectroscopy, the $C_0$ phase of the water–hydrogen mixture, produced at ∼400 MPa and recovered at room pressure and liquid nitrogen temperature, still containing a large fraction of molecular hydrogen. We demonstrate that, by means of a thermal

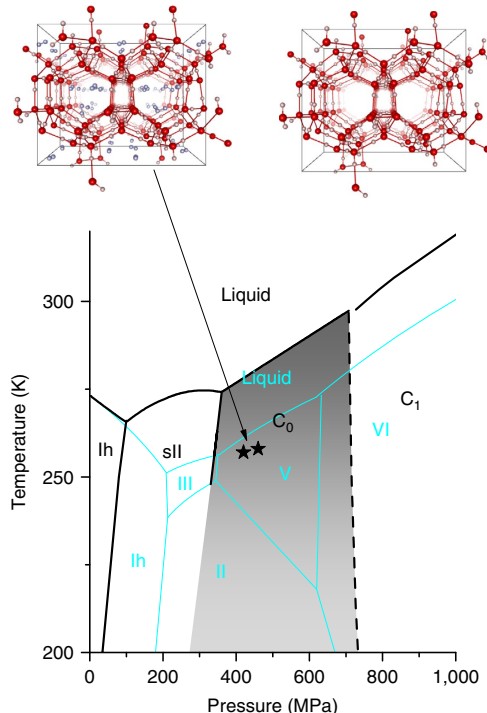

**Figure 1 | Stability region of the $C_0$ phase.** Projection on the $P - T$ plane of the phase diagram of the water–hydrogen mixture, for large $H_2$ concentration, in the region of interest. The solid black lines summarize the available experimental data[5,17–19,36–38] and do not have the same accuracy. The dashed black line shows the possible location of the $C_0$–$C_1$ transition[19]. For comparison, also the phase diagram of pure water is shown (thin cyan lines)[1,39]. Drawings of the molecular arrangement in the $C_0$ phase are pictured according to ref. 18 and to the results discussed later on, both with and without filling of hydrogen molecules, which are represented as dumbbell in a random orientation. The black stars represent the typical thermodynamic conditions for the synthesis of our samples.

treatment under vacuum, all the hydrogens can be removed, and that the new form of ice so obtained, named ice XVII, is metastable at room pressure below 120 K. The structure of the sample is checked before the removal of the hydrogen (that is, in the $C_0$ phase) and after it, showing that minimal changes, if any, are produced by the outgassing. The arrangement of water molecules in ice XVII gives rise to spiraling channels parallel to the crystallographic c axis, having a diameter of ∼6.10 Å, making this ice a porous material. As a matter of fact, we show in this work that ice XVII can adsorb and release hydrogen gas over and over, without evident change of structure.

## Results

**Sample production.** For this study we produce several batches of the solid $H_2$–$H_2O$ compound in the $C_0$ phase and recover the samples at room pressure and liquid nitrogen temperature, by application of a standardized and reproducible procedure. The structure of some specimens is examined by X-ray diffraction at the CRIST laboratory of the University of Firenze. The sample examined presents a considerable texture, which prevents a Rietveld analysis, but the measured diffraction pattern can be used to discriminate among several proposed structures. A Le Bail fit of the diffraction pattern is presented in Fig. 2, assuming the $C_0$-II structure, space group $P3_112$. Lattice constant are $a = 6.3313 \pm 0.0002$ Å and $c = 6.1058 \pm 0.0002$ Å. The other structures considered, namely sT′ ($P4_2/mnm$), Ih-$C_0$ ($Cc$)[21] and ice $I_c$ ($Fd\bar{3}m$), would give diffraction patterns in evident

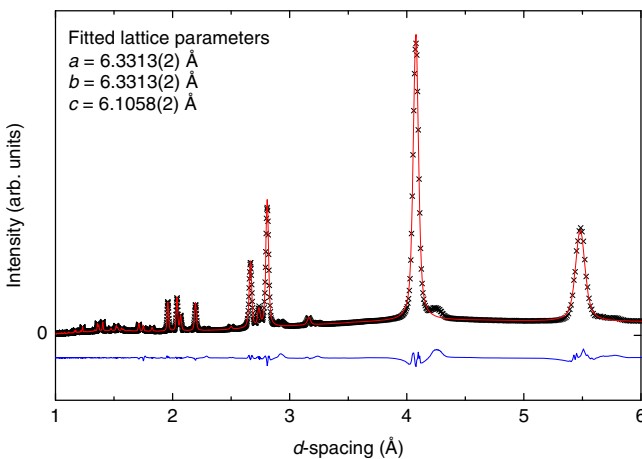

**Figure 2 | X-ray diffraction pattern.** Experimental X-ray diffraction pattern measured at ambient pressure and temperature of ∼100 K using Cu K$_\alpha$ radiation (black crosses). The red line represents the fit, performed with Le Bail method, to the experimental data using the $C_0$-II structure, space group $P3_112$. The value of the lattice parameters obtained from the fit procedure are reported in the figure. The blue curve is the difference between the experimental pattern and the fit.

disagreement with the experimental one. Details of the synthesis procedure and of the X-ray measurements are given in the Methods section.

**Measurement and interpretation of the Raman spectra.** By using our cryogenic Raman apparatus[22] we measure spectra during different $P-T$ cycles, observing the crystal lattice excitations and the rotations of the H$_2$ molecule (150–650 cm$^{-1}$), the OH stretching mode of the water molecule (3,000–3,400 cm$^{-1}$) and the vibron of the H$_2$ molecule (4,100–4,200 cm$^{-1}$). Our Raman spectra (see Fig. 3) and overall results add important information on the structure, which are consistent with only one of the proposed structures, namely the $C_0$-II one[18,20] with space group $P3_112$ depicted in Fig. 1. In this structure the water molecules form spiraling channels with a free bore hole along the $z$ axis of ∼5.26 Å and with a diameter of 6.10 Å, which can accommodate the H$_2$ molecules. Observing the spectra (Fig. 3a), we notice that the lattice phonon band (black line) has a broad smooth shape, showing similarities with the same band in both ice Ih (blue line) and sII clathrates (red line). The absence of sharp lines rules out the possibility that the water lattice in the $C_0$ phase might be proton ordered. This is at variance with the $C_1$-structure filled ice, which exhibits proton order and, consequently, a lattice phonon spectrum with evident sharp lines[17]. The rotational spectrum of the hydrogen molecules (Fig. 3c) presents well distinct lines, proving that the H$_2$ molecules rotate almost freely. The S$_0$(0) rotational band is split in probably three components, with a larger splitting than in clathrate hydrates. This is an indication of a more intense interaction of the hydrogen molecule with the environment and a larger perturbation of the rotational motion. According to the proposed structure, the hydrogen molecules are arranged in the channels, probably in a spiraling configuration[18], at a distance from water oxygen atoms of ∼3.1 Å and at ∼2.95 Å between each other, the same as in solid hydrogen at ∼1.5 GPa. The larger splitting of the rotational band components is accountable on the basis of the shortest distance between H$_2$ and H$_2$O molecules in the $P3_112$ structure, and indirectly confirms it. In addition, at such a short distance the H$_2$–H$_2$ anisotropic interaction may sustain collective rotational

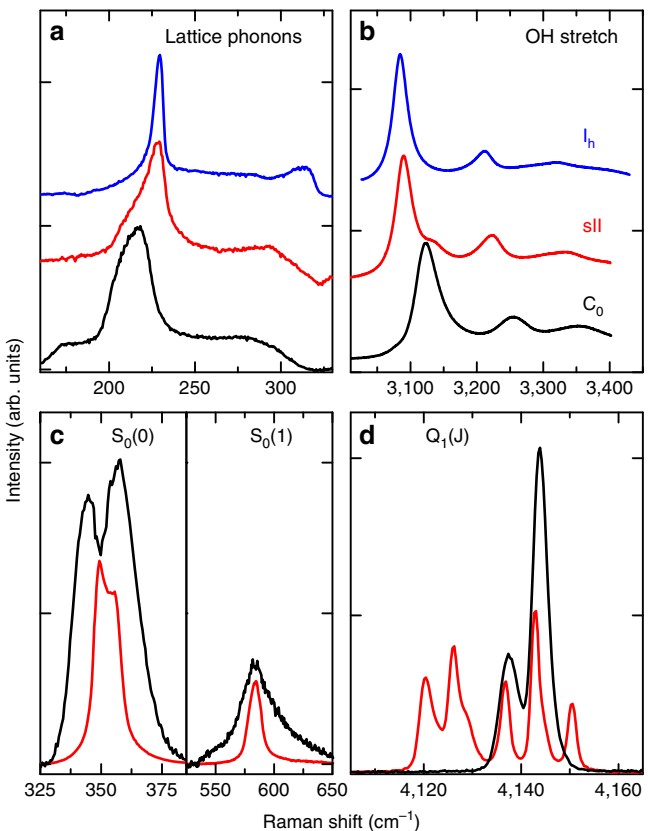

**Figure 3 | Raman spectra measured at 30 K.** Spectra measured for the $C_0$ sample (black lines), compared with those measured in sII clathrate[16] (red line) and ice Ih (blue line). The lattice phonon band in the range 120–310 cm$^{-1}$ (**a**) is at lower frequencies, whereas the OH stretching band (**b**) has the main peak at higher frequency with respect to ice and clathrates, as a manifestation of a larger O–O distance in the structure. The two rotational excitations, S$_0$(0) and S$_0$(1) of the hydrogen molecule, (**c**) are broader than in sII clathrates, indicating a stronger interaction with the host. A single pair of lines (Q$_1$(0) and Q$_1$(1)) observed for the H$_2$ vibron attests the existence of a single crystallographic site for the H$_2$ molecule in the $C_0$ crystal, opposite to the sII clathrate, where the different peaks correspond to different environments for the H$_2$ molecules (**d**).

excitations, as it happens in solid H$_2$ (ref. 23) and Ar(H$_2$)$_2$ high-pressure compound[24,25], which may contribute to the width of the rotational lines. The spectrum of the H$_2$ vibron region (Fig. 3d) consists of only one doublet (Q$_1$(0) and Q$_1$(1) lines), and this is in accordance with the occupation of a single crystallographic site for the hydrogen molecules, as in the proposed structure[18]. The OH stretching band (Fig. 3b) has the main peak at 3,123 cm$^{-1}$, a frequency higher than that of both sII clathrate and ice Ih. The OH vibrational frequency is known to decrease, while decreasing the O–O distance in different ices. This is observed both as a function of the considered compound[26] or, for the same structure, as a function of pressure, as reported for ice VII (ref. 27) and $C_1$-phase filled ice[28]. The higher frequency of the OH stretching mode observed for the $C_0$-structure filled ice is in accordance with the larger O–O distance in the structure $P3_112$ ($d_{OO} \simeq 2.83$–$2.85$ Å ) than in sII clathrate ($d_{OO} \simeq 2.77$–$2.79$ Å) and in ice Ih ($d_{OO} \simeq 2.75$ Å).

**Raman band intensities and hydrogen content.** We derive the hydrogen molar fraction $X$ from the intensity ratio of the hydrogen rotational lines and the lattice phonon band, $I_{rot}/I_{phon}$. This analysis is done, thanks to the assumption, that sounds

obvious, that the Raman intensity of a band is proportional to the number of molecules giving rise to it. This is anyhow an approximation, as the difference of polarizability of the same molecules in different environments may lead to tiny differences in intensity. This matter has been considered in section IIIa of ref. 16 and we refer to that study for a more thorough discussion. We calculate the calibration factor by fitting data arising from two independent experimental methods. A first set of data is obtained by using similar spectra of sII clathrates, for which the hydrogen content is calculated by counting the number of molecules in the large and small cages[16]. The spectra of these samples in the $200–900\,cm^{-1}$ range (not reported in ref. 16) are used to calculate $I_{rot}/I_{phon}$. To these data we add the information we derive from new volumetric measurements performed on our $C_0$-phase sample at three temperatures, namely 20, 50 and 80 K. The procedure is explained in detail in the Methods section. For the samples examined just after synthesis, whose typical spectra are represented in Fig. 3, we obtain $X \simeq 25\%$.

**Emptying of the $C_0$-structure filled ice**. We discover that the $C_0$ structure is metastable at room pressure even when emptied and is a new form of ice, namely ice XVII. The mechanical stability of the sample is tested by increasing the temperature, while keeping the sample under dynamic vacuum. We then observe the gradual release of the hydrogen from the sample, up to the complete undetectability of the hydrogen rotational lines obtained after pumping for 1–2 h at a temperature of $\sim 120$ K. Considering the sensitivity and the signal-to-noise ratio of our detection system, we can ascertain that the $H_2/H_2O$ molar fraction in the emptied sample is $<0.5\%$. During the heating process, we do not observe any abrupt change of the lattice phonon and OH stretching bands, demonstrating that no structural phase transitions have occurred. Decreasing again the temperature of ice XVII and comparing the spectra measured at the same low temperature, we notice after the annealing an increase in the frequency of

$\sim 7\,cm^{-1}$ for the lattice phonon bands (Fig. 4a) and a decrease of $\sim 20\,cm^{-1}$ for the OH mode (Fig. 4b). This indicates a decrease of the average O–O distance. The same indication we obtain from the concurrent increase of the lattice modes, indicating a stronger binding. However, the overall similarity of the band shape before and after annealing leads us to believe that the structure of the filled $C_0$ ice and that of ice XVII are essentially the same. The situation for the $C_0$ structure is different from previous observation in sII, where on cages emptying, the lattice constant and, consequently, the O–O distance increases[3]. The contraction of the water framework on inclusion of Ne in the cages of sII clathrate may be imputed to an attractive interaction between this atom and water. It is interesting to compare also with the situation of the He-hydrate, which become ice II on emptying[29]. Here the phenomenology is more complex, but the changes in the structure can be related to the mainly repulsive interaction between the He and O atoms. The precise nature of the effect of the annealing on the structure of the $C_0$ samples is not clear yet. The increased sharpness of the spectral features measured after the annealing suggests the decrease of defects in the structure. We recall that recently some authors[30], to explain the non-perfect refinement of the X-ray diffraction pattern with the $P3_112$ structure, have hypothesized that some nitrogen molecules, instead of water molecules, may have been trapped in the channels during the recovering of the sample in liquid nitrogen. A possible effect of the annealing might be the removal of these nitrogen molecules. To check this hypothesis, we measure Raman spectra of our samples, before and after this annealing process, also in the region of the $N_2$ stretching mode, observing, at 20 K before the annealing, a quite evident and sharp line at a frequency $2323.7 \pm 0.3\,cm^{-1}$. By virtue of their vibration frequency, sensibly lower than both that of $N_2$ gas $(2329.917\,cm^{-1})[31]$, and that of solid $N_2$ in the $\alpha$-phase (a doublet at 2327.5 and $2328.5\,cm^{-1})[32]$, we can establish that the $N_2$ molecules are trapped inside the structure. The Raman line of nitrogen molecules essentially disappears (it becomes 1,000 times

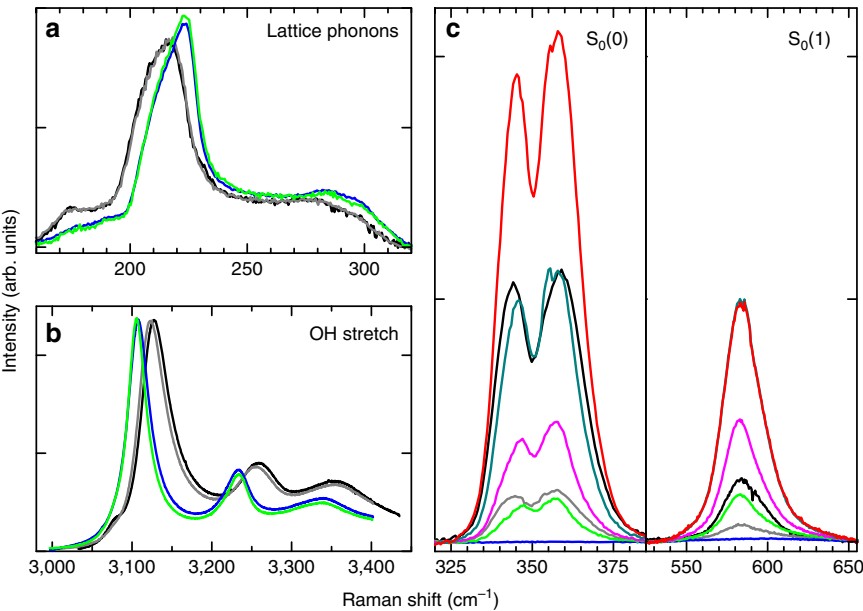

**Figure 4 | Effect of heating and hydrogen release.** The spectra of two different pristine samples measured at 30 K (black and grey lines in all panels) show some different filling (see **c**), probably due to unintentional heating during handling, but hardly any difference, or a very small one, in the lattice phonon (**a**) and OH stretching band (**b**) respectively. The empty $C_0$ ice, that is, ice XVII, cooled again at 20 K after annealing, (blue lines) show shifted and narrower spectra, because of increased bond strength between molecules and decrease of defects. Filling again ice XVII at 40 K does not alter much the spectra of the water molecules (green lines in **a**,**b**). Red lines in **c** show the spectra of the hydrogen molecules for the maximum filling obtained in this work $(X \simeq 40\%)$.

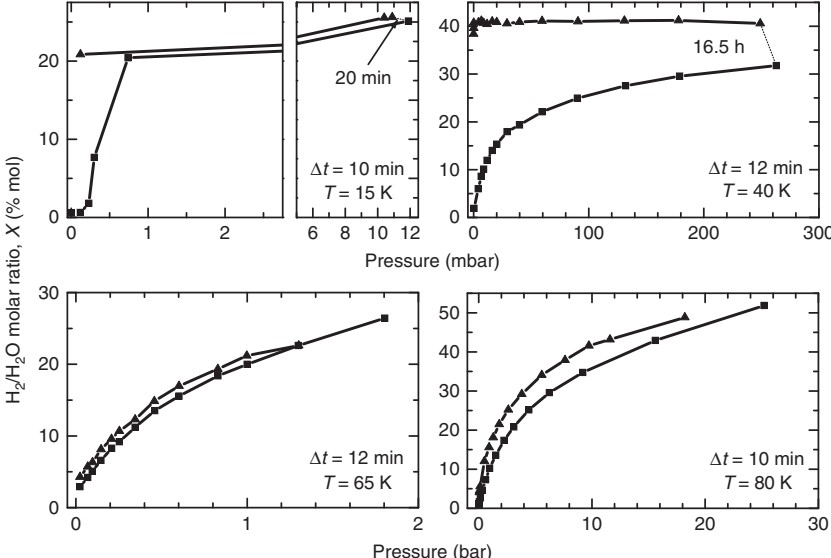

**Figure 5 | Adsorption isotherms measured at different temperatures.** We report the hydrogen molar fraction $X$ measured while increasing (square symbols) and decreasing (triangular symbols) hydrogen pressure in annealed ice XVII. Owing to calibration, $X$ has a systematic uncertainty of $\pm 10\%$. The time lag between pressure increase in the cell and Raman measurement in same series (same symbol) is indicated as $\Delta t$ in each panel, whereas we have joined with a thin dotted line measurements performed with larger delays. The kinetic effects that produce the large hysteresis observed at the two lower temperatures are absent or less evident increasing the temperature.

weaker) after the annealing process. The evident spectroscopic signatures of structural modification described above are a combined effects of the removal of both $H_2$ and $N_2$ from the sample. The structure of deuterated ice XVII is determined by means of neutron diffraction measurements performed on OSIRIS at ISIS, RAL (UK)[33]. Reitveld refinement of the data enables us to determine the structure. The empty structure is described by the space group $P6_122$, with oxygen and deuterium atoms in $6b$ and $12c$ positions, respectively. The space group used for the refinement has a higher symmetry than that proposed for the $C_0$-phase filled ice (ref. 20) on the basis of the data in ref. 18. This may be due to the presence in the channels of either $H_2$ guests and $N_2$ impurities. A complete description of the neutron diffraction experiment and fit procedure, together with the fitted structural parameters and water molecules geometry are reported in ref. 33.

**Refilling of ice XVII.** The unexpected property of ice XVII is that, when exposed to even a low pressure of hydrogen gas, it adsorbs the molecules up to an amount that is pressure dependent, but may grow larger than that initially present just after synthesis. As a matter of fact, we generally obtain $X = 25\%$ for pristine samples, possibly because of the presence of adsorbed nitrogen in the channel and/or of the handling of the sample for the insertion of it into the Raman apparatus. The adsorption process has initially a fast kinetics, even at 15 K. The rotational Raman spectra measured at 40 K after subsequent partial refilling steps are shown in Fig. 4c. Maximum gas pressure is in this case only 250 mbar, but the rotational intensity grows higher than that in the pristine $C_0$ sample. The shape of the rotational lines does not change significantly, demonstrating that $H_2$ is penetrating again in the same positions in the channels as those initially occupied at the time of synthesis. Normalizing the rotational Raman intensity with respect to the lattice phonon band and applying the calculated conversion constant, we estimate the $H_2$/$H_2O$ molar fraction. We notice that, in some instances, as for the most intense spectrum shown in Fig. 4c, the measured $H_2/H_2O$ molar ratio in the sample exceeds 40%, reaching higher values

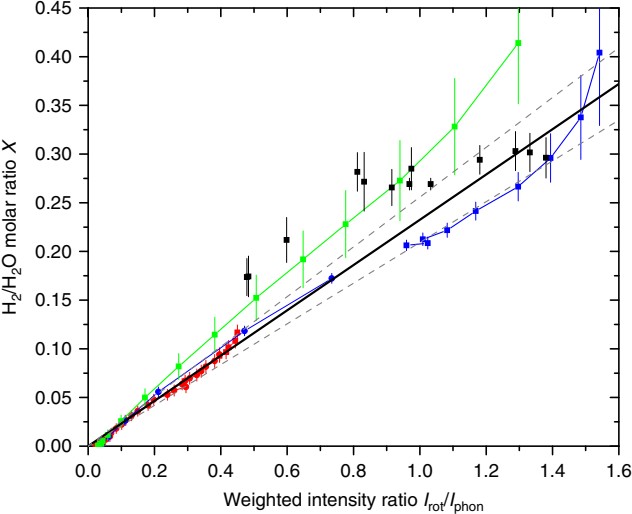

**Figure 6 | Raman intensity calibration.** Summary of the determinations of the $H_2/H_2O$ molar ratio plotted as a function of the concomitant measurements of the $I_{rot}/I_{phon}$ ratio, used to determine the calibration factor. The solid black squares represent measures of several instances of sII pure $H_2$ clathrate hydrates, synthesized at different pressures and/or undergone to different thermal cycles[16], whereas the other points joined by straight lines are volumetric measures performed in this work during the measurements of the 20, 50 and 80 K isotherms (red, green and blue, respectively). Different symbols (solid dots or squares) indicate the use of a different standard volume and/or a different transducer. The solid black line is the linear fit through zero, with slope 0.233 and the two dashed lines, with slopes $0.233 \pm 0.023$, represent the confidence interval. Error bars are estimates of maximum uncertainty as explained in the text.

than the established theoretical maximum ($48/136 \simeq 35\%$) for sII hydrates.

We measure several adsorption isotherms, deriving the amount of adsorbed $H_2$ as a function of pressure, at several different temperatures between 15 and 80 K. Results for four temperatures

are presented in Fig. 5. Pressure is increased gradually (squares in Fig. 5) and, when possible, measurements are taken after the same delay $\Delta t$ from pressure increase, which is indicated in each panel. The pressure at which saturation is reached is strongly dependent on temperature and ranges between a few millibar at 15 K to several bar at 80 K. By measuring with the same method, we also examine the gas desorption, while releasing pressure. A large hysteresis and other kinetics effects are evident at the lower temperatures. At 15 K, after decreasing pressure down to zero, the sample does not sensibly release hydrogen in $\Delta t = 10$ min. The same effect is observed at 40 K, the filling remaining almost constant for at least 36 min. At this temperature, the increase in the adsorbed gas observed at $\sim 240$ mbar, after a 16.5 h-long exposure to hydrogen, indicates that at least two time scales dominate the phenomenon: one fast adsorption in the outermost sites of the channels and a slower one, limited by molecular diffusion, inside the channels. At higher temperatures, 65 and 80 K, we observe a very small hysteresis, indicating that the time needed to hydrogen to diffuse for distance of the order of the linear dimension of the sample grains is quite shorter than the time interval between measurements. At temperature above 50 K, the desorption process is quick (compared with sII clathrates). This is not surprising at all, as the channels in ice XVII are much wider than the free space inside both the six-membered and five-membered rings. From the isotherm hydrogen uptake as a function of the pressure presented in Fig. 5, it is possible to estimate, by means of the Van't Hoff equation, the enthalpy of adsorption, $\Delta h$, for a given hydrogen uptake $X$:

$$\frac{\Delta h}{R} = \left.\frac{\partial \ln(p/p_0)}{\partial 1/T}\right|_X \qquad (1)$$

where $R$ is the gas constant[34,35]. For this calculation, we use the data collected at 50, 65 and 80 K, estimating the derivative with a straight line fit. The resulting values of $-\Delta h$ decrease with increasing molar ratio $X$ starting from $\sim 5$ kJ mol$^{-1}$ at $X \simeq 10\%$ down to $\sim 2$ kJ mol$^{-1}$ at $X \simeq 40\%$, in agreement with typical values for physisorption.

## Discussion

This work is the first characterization of the dynamics of the $C_0$-phase filled ice. Moreover, two remarkable results arise from this study. First, we discover that the $C_0$ structure is metastable when emptied. Owing to its large range of stability (up to 120 K at least), it should be counted as a new form of ice and named ice XVII. Second, we find that ice XVII is capable to adsorb hydrogen gas in large quantities (presumably, on the basis of its structure up to $X = 50\%$), without substantial change in its structure. We believe that, more than the maximum estimated hydrogen uptake, it is the fast reversibility of gas adsorption, the theoretically infinite number of possible cycles (there is no chemical change of the substance involved) and the relatively modest pressures at which the process occurs, at temperature close to that of liquid nitrogen, which make ice XVII really appealing for hydrogen storage applications.

## Methods

**Procedure for the synthesis of the sample.** We produce several instances of solid $H_2$–$H_2O$ compound by application of a standardized procedure. The check of the recovered samples with Raman scattering confirms that we have always obtain the sample in the same $C_0$-phase. To synthesize the samples, we introduce a few grams of finely ground ice into a beryllium-copper autoclave and we expose the sample to hydrogen gas at a pressure in the range or above 430 MPa, at a temperature of $\sim 255$ K ($-18\,^\circ$C). We prudentially leave the sample under pressure for a few days, even though, while rising pressure during synthesis, we observe at $\sim 360$ MPa an abrupt decrease of the rate of pressure rise, which is a clear indication of large hydrogen adsorption by the ice. Next, we quench the autoclave in liquid nitrogen when still under pressure and, finally, after releasing

the pressure, we recover the sample at liquid nitrogen temperature, in the form of a fine powder. The same procedure is applied for the synthesis of the samples to be investigated by X-ray diffraction, except that in this case a drop of water is frozen from the beginning inside a few common X-ray glass capillaries, having 0.5 mm diameter and a length of $\sim 10$–15 mm. The frozen water occupies a capillary length of $\sim 2$ mm. The capillaries are inserted in the autoclave where they undergo the same standardized procedure and are then recovered and handled at liquid nitrogen temperature.

For Raman measurements, a small amount of sample is inserted in our optical cell, in contact with the cold finger of a closed-cycle He cryostat[22]. The transfer process has to be accomplished at almost liquid nitrogen temperature and in a dry-nitrogen atmosphere, exerting particular care to avoid sample heating. Once the Raman cell is filled with the sample, it is sealed with the optical window and purged with helium gas. The sample cell can then be set to any temperature between room temperature and $\simeq 10$ K. Gas pressure in the cell can be varied and measured with an accuracy of $\sim 1\%$.

**Raman and X-ray measurements.** Raman spectra are excited by means of an Ar ion laser at 514.5 nm, with a power of $\sim 30$ mW on the sample, focused on a spot of about 30 μm. Scattered light is collected in an almost backscattering geometry, focused on the entrance slit of a Spex spectrograph and recorded by a cooled CCD (charge-coupled device) detector (Andor). The maximum resolution of the instrument is 0.4 cm$^{-1}$. Similar care has to be exerted to mount the capillary on the X-ray diffractometer (four circle Oxford Diffraction XcaliburPX). The capillary is maintained at low temperature ($\sim 90$–100 K) by a cold nitrogen blow provided by a Oxford Cryosteam system. Diffractograms are collected using a copper X-ray source and recorded by a 165 mm diameter CCD detector. Total collection time is of the order of 7 min. During collection, the capillary is turning rapidly.

**Calibration of the Raman intensity.** We measure the molar fraction of hydrogen molecules contained in the samples of filled ice by computing the ratio of the intensity of the $S_0(0)$ and $S_0(1)$ rotational lines, weighted by the inverse of the respective cross-section with the intensity of the lattice phonon band. Specifically, if we indicate with $\sigma_0$ and $\sigma_1$ the Raman cross-section for the $S_0(0)$ and $S_0(1)$ rotational lines, and with $I_0$ and $I_1$ their intensity, respectively, the quantity $I_{rot} = I_0/\sigma_0 + I_1/\sigma_1$ is proportional to the number of $H_2$ molecules in the lowest two $J = 0$ and $J = 1$ rotational states, that is, at low temperature, to the number of $H_2$ molecules giving Raman scattering. Dividing $I_{rot}$ by the intensity of the water lattice phonon band, $I_{phon}$, we obtain a quantity which is independent of laser intensity and detector efficiency, and proportional to the $H_2/H_2O$ molar ratio in the sample. This procedure needs a calibration, which we have accomplished fitting data obtained with two independent methods. For several samples of sII clathrates studied in our laboratory, the hydrogen content has been indirectly derived, by counting the molecules in the small and large cages[16], whereas the spectra of the same samples in the 200–900 cm$^{-1}$ range are in our records, even if not reported in ref. 16. This allows us to draw some calibration points (black squares) in Fig. 6. We have performed additional volumetric measurements on our $C_0$ sample, while measuring, with Raman scattering, the adsorption isotherms at three temperatures, namely 20, 50 and 80 K. These results are reported as coloured dots in Fig. 6. The estimated uncertainty on these experimental points depends critically on the precision of the different calibrated volumes, of the pressure transducer used and on the pressure of the measurement. The cumulative linear fit gives a calibration factor of $0.233 \pm 0.023$, which is the value we have used in this work.

**Data availability.** The data that support the findings of this study are available from the corresponding author upon reasonable request.

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

## Acknowledgements

We thank all the personnel at the Centro di Cristallografia of the Università degli Studi di Firenze (CRIST) for their assistance in the X-ray measurements. We are also grateful for the support of Francesco Grazzi (CNR-ISC, Firenze) in the analysis of the X-ray and neutron diffraction data.

## Author contributions

L.d.R. performed the high pressure synthesis of the sample, the low-temperature Raman measurements and the X-rays and neutron diffraction measurements. M.C. set up the high-pressure apparatus for the sample synthesis, performed the synthesis of the samples for Raman scattering, X-ray and neutron diffraction, and executed the X-ray and neutron diffraction measurements. L.U. designed the study, discussed data collection and analysis and wrote the paper.

## Additional information

**Competing financial interests:** The authors declare no competing financial interests.

**Publisher's note**: 

