## [Peer Review File · Nature Communications]

Reviewers' comments:

Reviewer #1 (Remarks to the Author):

In this work the authors have pressurized ice with hydrogen gas to form a filled ice 'C0' phase and have investigated the compound using Raman spectroscopy at different PT conditions. The work is quite interesting in the sense that H₂ can apparently be removed by heating (and /or pumping), possibly creating a new form of metastable ice, akin to the recent work of Kuhs and coworkers on type II clathrate hydrate. In addition, H₂ can re-adsorb suggesting potential application in the area of H₂ storage. The general idea is a very good one and the authors present lovely spectra, however, the evidence presented is not sufficient to substantiate the entirety of claims made. I would suggest publication pending additional experimental results.

The main downfall of the work is that no structural evidence is provided. Upon inspection of the literature, only refs. 5-6 have produced this 'C0' phase, but no accurate structural description is even available! The model used here (proposed in ref. 5) has completely unphysical O-O distances. Later, Sminov and Stegailov (J. Phys. Chem. Lett. 4, 3560 (2013) -- a reference that was missed here), calculated variants of the 'C0' structure from ref. 5 as well as quartz and the sT' structure. Based on their results, they suggested 'C0-II' and sT' both as viable candidates. After this, Oganov and coworkers (ref. 21) used DFT-based structure searching to predict two different flavors of 'C0' phase, one whose basic structural framework is similar to the model of ref.5, but these searching methods are restricted in the number of atoms, so any larger clathrate-sized unit cells can easily be missed. This discussion in the paper is clearly lacking and the claim of a new form of ice seems a bit of a stretch with no structural evidence. Perhaps it is correct, but how can one be certain that the 'C0' phase was even produced in this work? Other fairly recent work by Kumar et al. (Angew. Chem., 51, 1, (2012)) shows hydrogen encapsulation in ice Ic. One cannot fully exclude the formation of another phase without clear structural information. With that being said, assuming that the phase formed is indeed 'C0', the new insights from Raman observations do help place constraints which will be helpful to resolve the structure.

The H₂:H₂O molar ratio used throughout this work is derived completely from Raman measurements. The only basis for this appears to be stated as "we have calculated the same intensity ratio for a large number of spectra of sII clathrates, for which the hydrogen content is known (ref. 18)." But upon reviewing Ref. 18, it seems that no actual direct measurements of H₂:H₂O ratios were made! How can one be certain that the technique is accurate? How is the reported accuracy of 15% justified? Qualitatively, I take no issue with the trends observed, but I see no convincing evidence that these qualitative trends can be transformed into quantitative ratios. Is there another standard that could be used to benchmark the results? What about direct volumetric measurements? This would also be most helpful for resolving the structure.

The hydrogen storage angle seems a bit tenuous. The prospect of a new metastable water framework and insights to the guest-host interactions is sufficient justification for this work. Perhaps H₂ storage could be of some interest, but it's probably not feasible. I suggest toning this aspect down else really showing quantitative results and comparing with alternatives at similar thermodynamic conditions.

The use of 'stable' / 'stability' is unclear. If true, the empty phase is NOT a thermodynamically stable phase. It's a metastable one that can only exist at low T due to a specific kinetic pathway.

Page 2: Change "This latter" to "The latter".

With regards to the Raman peak widths it is unclear how this necessarily dictates stronger

perturbations or increased interactions. The central frequencies are not so far perturbed in energy. Perhaps a slight increase in splitting of the rotational energy levels, but not drastic compared with the clathrate. Perhaps increased disorder can account for the broadening. Also, the normalization (to water phonon?) is unclear for discussion of peak broadening.

Fig 1 is useful to see the conditions of the experiment, but it should be noted that the phase boundaries are only estimates and have not been measured rigorously. The work of Vos and Dyadin should be included on this figure.

Adsorb and absorb are used interchangeably in the paper.

In summary, this is an excellent idea, but the work seems too preliminary for publication in Nature Communications at present.

Reviewer #2 (Remarks to the Author):

The authors report on Raman measurements of the hydrogen release from the cubic C0 structure of hydrogen filled ice at different temperatures at ambient pressure. They show that the empty C0 structure can be produced under vacuum and the emptied crystal can absorb again and release hydrogen in a fraction depending on the temperature.

This is an interesting and clearly written work, which possibly can qualify for Nature Communications if the following issues are properly addressed:

1) The authors state that they "calibrate" the Raman quantification factors using the H₂-filled sII clathrate structure and claim that they know the absolute cage occupancies for this phase. They refer to their own work (reference 18). It is unclear to me how one can get absolute values from Raman only - as one only gets relative occupancies. The authors should better explain how they estimate the H₂ amount from the rotational peaks of the molecule and why, at difference with Ref 18, they don't estimate it from the H₂ stretching peak. The paper of A. Lokshin, et al., Phys. Rev. Lett. 2004 should also be quoted in this discussion. A further tricky question concerning this estimation is whether the Raman cross-sections for singly, doubly etc occupied cages is really the same.

2) The emptying proceeds quickly (compared to sII clathrates), yet this is not entirely surprising as the channels are wider are the bottlenecks (in particular the 5-membered rings). Still, I miss a discussion of this observation (with respect to the Falenty et al. 2014 paper) and in particular an estimate of the activation energy of the out-diffusion from the T-dependent data presented in Fig.4.

3) Likewise, the authors compare in a cursory way the diffusion constants estimated in reference 28. How can they do that if they do not know the particle size of their powders? Or how else did they estimate this quantity?

4) The authors observe a frequency decrease of the O-H band upon emptying indicating a decrease in the H-bonded O...O distance. But what do they mean with "evolution towards a more stable molecular arrangement"? In any case, the situation for the C0 structure is different from previous observation in sII, where upon emptying the H-bonded distance increases. A discussion of this difference should be attempted - at least this different behavior should be mentioned.

In this context it may also be well worthwhile to look at the empty ice II structure as compared to the He-filled one (Lobban et al. 2002 JCP117:3928-3934), from which a more complex picture emerges: Some O...O distances increase some others shrink upon emptying, the flat ring diameter decreases upon emptying while the puckered ring diameter stays the same. I expect a somewhat similar complexity also for the channel-like C0 structure, yet this is difficult to obtain from Raman. The authors should realize this complexity and moderate their statements accordingly.

5) The authors attribute the uptake of hydrogen during refilling to the fact that some N₂ trapped in the C0 channel could have been released after the first emptying exercise. Do the authors see the N₂ vibron in their initial Raman spectra?

6) In terms of the claimed H₂-storage applications they should provide more information as for which fields it could be useful. The vague statement that the structure upon reloading is more filled than initially needs to be backed up. Moreover, if one loads it at liq.N₂ temperature at whatever pressure as soon as you take the pressure off it will empty again - the question thus is how to avoid bleeding out during storage. There is no energetic gain if you have to cool to liq. N₂ temperatures.

7) Authors discuss hydrogen molar ratio with respect to water. Actually I believe that when discussing about H₂-storage possibilities it is more ground discussing about H₂ mass ratio, as in W.L. Mao, PNAS 20, 2004. This last paper should also be quoted.

Reviewer #3 (Remarks to the Author):

The paper reports the production of a solid H₂-H₂O compound in the C0 phase and recovered the sample at room pressure and liquid nitrogen temperature. The H₂ rotational and OH and H₂ stretching vibrational spectra were used to monitor the presence of H₂. XRD was also used to probe the positions of the oxygen atoms in the structure. The Raman spectra and overall results are consistent with the proposed trigonal structure (space group P3112), The C0 structure appears to be stable even when emptied. The structure of the filled and empty C0 ice are essentially the same.

Several adsorption isotherms, deriving the amount of adsorbed H₂ as a function of pressure, at several different temperatures between 15 to 80 K are presented in Figure 4. A large hysteresis and other kinetics effects are evident at the lower temperatures.

There is some interesting science in the paper. However, the isotherm evidence for a potential link to H₂ storage is quite weak. This potential application was quoted in the abstract 'Given these striking properties, we believe that ice XVII may be of large interest also for application in hydrogen storage, much more than what hydrogen clathrate hydrate have been in the past decade.' The isotherm data in Figure 4 are only given at low temperature (15-80K). Therefore the current evidence presented is weak. Ambient temperature isotherm measurements are required for justification of H₂ storage applications together with deliverable H₂ capacity.

Reviewers' comments:

Reviewer #1 (Remarks to the Author):

The revised manuscript is a marked improvement over the original and most of the original issues have now been addressed. Because the authors are claiming a new phase of ice, they should also show XRD measurements after hydrogen has been removed from the sample. Once this is complete I recommend that the work be published (assuming that there is no structural change).

Minor note: the discussion state that the structure is "stable when emptied," but should say metastable, or dynamically stable.

Reviewer #2 (Remarks to the Author):

In their revised manuscript entitled "A new porous water ice stable at atmospheric pressure obtained by emptying a hydrogen filled ice" authors have:

1. included the results of a x-ray diffraction check of one of their samples in the phase C0, where the measured diffraction pattern is fitted, by the Le Bail method, with the one resulting from the structure named C0-II, with good match. This gives now a more convincing indication of the structure of the sample which was not provided with Raman data only.
2. included the results of new measurements of three absorption isotherm, where they determined, by means of a volumetric method, the quantity of adsorbed hydrogen at each step, to compare it with the intensity of the measured Raman hydrogen spectrum. These data lead now to a calibration based on two independent methods, reported and discussed in the Methods section, which is more convincing.
3. added a qualitative discussion of these kinetic effects and given a reasonable estimate of the isosteric adsorption enthalpy.
4. checked with Raman the presence of Nitrogen trapped in the C0 channel
5. mitigated the importance of this material for hydrogen storage.

Addressing these points they replayed in a convincing way to my main concerns, I thus judge the present version of the manuscript qualified for publication in Nature Communications.

REVIEWERS' COMMENTS:

Reviewer #1 (Remarks to the Author):

The MS should be accepted.

Reviewers' comments:

Reviewer #1 (Remarks to the Author):

In this work the authors have pressurized ice with hydrogen gas to form a filled ice 'CO' phase and have investigated the compound using Raman spectroscopy at different PT conditions. The work is quite interesting in the sense that H₂ can apparently be removed by heating (and /or pumping), possibly creating a new form of metastable ice, akin to the recent work of Kuhs and coworkers on type II clathrate hydrate. In addition, H₂ can re-adsorb suggesting potential application in the area of H₂ storage. The general idea is a very good one and the authors present lovely spectra, however, the evidence presented is not sufficient to substantiate the entirety of claims made. I would suggest publication pending additional experimental results.

The main downfall of the work is that no structural evidence is provided. Upon inspection of the literature, only refs. 5-6 have produced this 'CO' phase, but no accurate structural description is even available! The model used here (proposed in ref. 5) has completely unphysical O-O distances. Later, Sminov and Stegailov (J. Phys. Chem. Lett. 4, 3560 (2013) -- a reference that was missed here), calculated variants of the 'CO' structure from ref. 5 as well as quartz and the sT' structure. Based on their results, they suggested 'CO-II' and sT' both as viable candidates. After this, Oganov and coworkers (ref. 21) used DFT-based structure searching to predict two different flavors of 'CO' phase, one whose basic structural framework is similar to the model of ref.5, but these searching methods are restricted in the number of atoms, so any larger clathrate-sized unit cells can easily be missed. This discussion in the paper is clearly lacking and the claim of a new form of ice seems a bit of a stretch with no structural evidence. Perhaps it is correct, but how can one be certain that the

'CO' phase was even produced in this work? Other fairly recent work by Kumar et al. (*Angew. Chem.*, 51, 1, (2012)) shows hydrogen encapsulation in ice Ic. One cannot fully exclude the formation of another phase without clear structural information. With that being said, assuming that the phase formed is indeed 'CO', the new insights from Raman observations do help place constraints which will be helpful to resolve the structure.

We have added the information the reviewer is asking. The discussion of the structures proposed in the literature for the CO phase, that was extremely scarce in the previous version due to length requirements, is now, in our opinion, exhaustive. We were aware of the work of Smirnov and Stegailov, although not mentioned in the bibliography (now ref. 24), and the structures taken into account in that paper (CO-I, CO-II, sT', alpha-quartz etc) were checked. We are aware also, as the reviewer remarks, that the CO-I structure do not allow correct hydrogen bonding between water molecules. This is true also for the alpha-quartz structure with the parameters listed in ref. 24, while the structures indicated with CO-II and sT' (the latter after correction of a typographical error in the paper) do.

In the revised version of our paper we have included the results of a structural check, done by means of x-rays, on one of our batch in the phase CO, adding a figure where the measured diffraction pattern is fitted, by the Le Bail method, with the one resulting from the structure named CO-II, with good match. All the other proposed structures would produce diffraction pattern in strong contrast with the measured one.

The H₂:H₂O molar ratio used throughout this work is derived completely from Raman measurements. The only basis for this appears to be stated as "we have calculated the same intensity ratio for a large number of spectra of sII clathrates, for which the hydrogen content is known (ref. 18)." But upon reviewing Ref. 18, it seems that no actual direct measurements of H₂:H₂O ratios were made! How can one be certain that the technique is accurate? How is the reported accuracy of 15% justified? Qualitatively, I take no issue with the trends observed, but I see no convincing evidence that these qualitative trends can be transformed into quantitative ratios. Is there another standard that could be used to benchmark the results? What about direct volumetric measurements? This would also be most helpful for resolving the structure.

The method to derive the H₂ molar concentration from the intensity ratio of the hydrogen Raman rotational lines with the water lattice band needs a calibration. In the paper originally submitted this calibration was obtained only on the basis of previous Raman measurements on hydrogen clathrates. We agree with the reviewer that the procedure was not explained clearly. In this revised version of the paper we not only clarify this issue, but we include the results of new measurements of three more absorption isotherm, during which we have recorded, by means of a volumetric method, the quantity of adsorbed hydrogen at each step, to compare it with the intensity of the measured Raman hydrogen spectrum. These data all together lead to a calibration based on two independent methods, which is reported and discussed in the Methods section and in fig. 6. To specifically answer to this remark, we want to clarify how the data of ref. 18 (now ref. 16) are used for this calibration (even though the volumetric determinations are more accurate). The reviewer is correct to state that in ref. 18/16 no *DIRECT* measurements of H₂:H₂O ratios were made, but this ratio is nevertheless derived by counting the average number of hydrogen molecules in the large and small cages (see table III in ref. 18/16) with the knowledge of the number of cages and water molecules per unit cell. What is not reported in ref. 18/16 is the spectrum in the water lattice/hydrogen rotation region, by means of which one can calculate the intensity ratio. These spectra for exactly the same samples are however in our possess, and have

been used to extract the intensity ratio, used to report the black squares Fig. 6. The accuracy estimate (now 10%) derives from the spread of the points in fig. 6.

The hydrogen storage angle seems a bit tenuous. The prospect of a new metastable water framework and insights to the guest-host interactions is sufficient justification for this work. Perhaps H₂ storage could be of some interest, but it's probably not feasible. I suggest toning this aspect down else really showing quantitative results and comparing with alternatives at similar thermodynamic conditions.

We agree with the reviewer that possible use of this material for hydrogen storage purpose is not the main argument of this work, and mention to it has been removed from the abstract. Some comments on this subject remains in the conclusions. A quantitative evaluation of the energy balance for its practical use as hydrogen storage material will be attempted, but it is beyond the scope of this work.

The use of 'stable' / 'stability' is unclear. If true, the empty phase is NOT a thermodynamically stable phase. It's a metastable one that can only exist at low T due to a specific kinetic pathway.

This observation is correct. In the revised text, we have paid attention on the use of these terms.

Page 2: Change "This latter" to "The latter".

This correction has been done.

With regards to the Raman peak widths it is unclear how this necessarily dictates stronger perturbations or increased interactions. The central frequencies are not so far perturbed in energy. Perhaps a slight increase in splitting of the rotational energy levels, but not drastic compared with the clathrate. Perhaps increased disorder can account for the broadening. Also, the normalization (to water phonon?) is unclear for discussion of peak broadening.

The rotational Raman spectra are reported in Fig. 3c, and their peak shape and width are discussed qualitatively at pag. 4. The S(0) rotational band is split in probably three components, (as in clathrate hydrates) but the splitting is larger. This splitting is due in a first approximation to the anisotropic components (with respect to H₂ orientation) of the local field acting on one H₂ molecule, that is therefore stronger in the CO phase than in clathrate. This was in our view the meaning of "stronger perturbation of the rotational motion". This sentence has been slightly changed to clarify this concept.

Each rotational component is broadened, as in clathrates, positively by the (proton) disorder. In our view the rotational Raman band would be narrower if the H₂ molecule were trapped (alone) in a cage such that present in the proposed sT' structure.

The normalization of the bands reported in fig. 3c is arbitrary and is done to avoid superposition of the lines. One can, nevertheless, readily estimate the FWHM of each band.

Fig 1 is useful to see the conditions of the experiment, but it should be noted that the phase boundaries are only estimates and have not been measured rigorously. The work of Vos and Dyadin should be included on this figure.

We agree with the observation that in this figure the presence or absence of experimental points was not coherent, and that some phase boundaries are only estimates and have not been measured accurately. On the other hand, this figure has only an indicative intent, to recall the reader the situation, and to show the conditions of the synthesis of our samples. We have therefore removed the experimental points and referred to the original papers where these phase lines, with different accuracy, have been measured or hypothesized. In this respect, the C0-C1 boundary has been drawn as a dashed line.

Adsorb and absorb are used interchangeably in the paper.

We have corrected this terminology.

In summary, this is an excellent idea, but the work seems too preliminary for publication in Nature Communications at present.

Reviewer #2 (Remarks to the Author):

The authors report on Raman measurements of the hydrogen release from the cubic C0 structure of hydrogen filled ice at different temperatures at ambient pressure. They show that the empty C0 structure can be produced under vacuum and the emptied crystal can absorb again and release hydrogen in a fraction depending on the temperature.

This is an interesting and clearly written work, which possibly can qualify for Nature Communications if the following issues are properly addressed:

1)The authors state that they "calibrate" the Raman quantification factors using the H2-filled sII clathrate structure and claim that they know the absolute cage occupancies for this phase. They

refer to their own work (reference 18). It is unclear to me how one can get absolute values from Raman only - as one only gets relative occupancies. The authors should better explain how they estimate the H₂ amount from the rotational peaks of the molecule and why, at difference with Ref 18, they don't estimate it from the H₂ stretching peak. The paper of A. Lokshin, et al., Phys. Rev. Lett. 2004 should also be quoted in this discussion. A further tricky question concerning this estimation is whether the Raman cross-sections for singly, doubly etc occupied cages is really the same.

The discussion of the calibration of the Raman intensity ratio (H₂/H₂O) is now completely developed in the Methods section. Some questions the reviewer is asking are explained in the answer to a similar question of reviewer #1. The estimation of the H₂/H₂O molar ratio was derived in ref. 18 (now ref. 16) from the stretching mode since this mode has a different frequency for different molecular occupation of the cages. Therefore it was possible to count the average number of hydrogen molecules in the large and small cages (see table III in ref. 18/16). This number, with the knowledge of the number of cages and water molecules per unit cell, gives the hydrogen molar concentration. The same method is not possible here since the vibrational spectrum is only a doublet and does not carry this information. In this revised version of the paper we include the results of new measurements of three more absorption isotherms, during which we have recorded, by means of a volumetric method, the quantity of adsorbed hydrogen at each step, to compare it with the intensity of the measured Raman hydrogen spectrum. These data all together lead to a more precise (10 %) calibration based on data obtained by two independent methods, where the volumetric ones are the most accurate.

Concerning the variation of the hydrogen molecular cross section as a function of the environment, this issue has been discussed in ref.18/16, sec IIIa, and those conclusions apply also here. In that paper we have considered local field effects (that cancel out when intensity ratio is computed) and interaction-induced extra components of the hydrogen molecular polarizability, whose effect can be estimated to be as small as few parts over ten thousand.

2) The emptying proceeds quickly (compared to all clathrates), yet this is not entirely surprising as the channels are wider are the bottlenecks (in particular the 5-membered rings). Still, I miss a discussion of this observation (with respect to the Falenty et al. 2014 paper) and in particular an estimate of the activation energy of the out-diffusion from the T-dependent data presented in Fig.4.

A qualitative discussion of these kinetic effects is now included at the end of the paper, before the Discussion section, where we report an estimate of the isosteric adsorption enthalpy. This value is quite small (only about 2-5 kJ/mol). In the hypothesis that the adsorption activation energy, E_a, is much smaller than desorption activation energy E_d, the enthalpy of adsorption is equal to -E_d. The small value of E_d justifies the rapid desorption of hydrogen.

3) Likewise, the authors compare in a cursory way the diffusion constants estimated in reference 28. How can they do that if they do not know the particle size of their powders? Or how else did they estimate this quantity?

The comparison was made assuming an average grain size of about 200 micron. However, given the uncertainty in this value, in the diffusion coefficient (computed only at 140 K) and in the

dependence of this coefficient on the temperature, this comparison risks to be meaningless. We have removed this sentence from the paper.

4) The authors observe a frequency decrease of the O-H band upon emptying indicating a decrease in the H-bonded O...O distance. But what do they mean with "evolution towards a more stable molecular arrangement"? In any case, the situation for the C0 structure is different from previous observation in sII, where upon emptying the H-bonded distance increases. A discussion of this difference should be attempted - at least this different behavior should be mentioned.

In this context it may also be well worthwhile to look at the empty ice II structure as compared to the He-filled one (Lobban et al. 2002 JCP117:3928-3934), from which a more complex picture emerges: Some O...O distances increase some others shrink upon emptying, the flat ring diameter decreases upon emptying while the puckered ring diameter stays the same. I expect a somewhat similar complexity also for the channel-like C0 structure, yet this is difficult to obtain from Raman. The authors should realize this complexity and moderate their statements accordingly.

We thank the reviewer for bringing to our attention the structural aspects discussed in the paper by Lobban. Some lines to comment on this have been added to the text. In connection to the initial presence of some nitrogen molecules inside the structure, we have moderated our statements on the structural effects of the emptying process. A more precise determination of these structural changes would require diffraction measurements during the first annealing process and subsequent hydrogen adsorption and release.

5) The authors attribute the upptake of hydrogen during refilling to the fact that some N₂ trapped in the C0 channel could have been released after the first emptying exercise. Do the authors see the N₂ vibron in their initial Raman spectra?

In the time elapsed between the submission of the first and the present versions of the paper, we have been able to check with Raman the presence of Nitrogen, with definitive positive results. The results are described in the text.

6) In terms of the claimed H₂-storage applications they should provide more information as for which fields it could be useful. The vague statement that the structure upon reloading is more filled than initially needs to be backed up. Moreover, if one loads it at liq.N₂ temperature at whatever pressure as soon as you take the pressure off it will empty again - the question thus is how to avoid bleeding out during storage. There is no energetic gain if you have to cool to liq. N₂ temperatures.

The sample, when refilled after the first annealing, adsorbs more H₂ than initially present (generally $X=25\%$), as evidenced by the adsorption isotherms in Fig. 5. A reason for that may be the release of adsorbed nitrogen molecules. A sentence to clarify this matter has been added at the beginning of the subsection **Refilling of ice XVII**. The importance of this material for hydrogen storage is not the main argument of this work, and has been further mitigated in this new version. The existence of a new form of ice, ice XVII, metastable at ambient pressure, which quickly adsorbs

and releases gases seems to us a sufficient justification for the publication of this paper. A quantitative evaluation of the energy balance for application of ice XVII as hydrogen storage material will be attempted, but it is beyond the scope of this work.

7) Authors discuss hydrogen molar ratio with respect to water. Actually I believe that when discussing about H₂-storage possibilities it is more ground discussing about H₂ mass ratio, as in W.L. Mao, PNAS 20, 2004. This last paper should also be quoted.

H₂:H₂O molar ration and H₂:(H₂+H₂O) mass ratio are directly determinable from each other. Given the fact that, also following the request of the reviewer#1, in this version we have mitigated the importance of this material for hydrogen storage, we do not consider important this change of units.

Reviewer #3 (Remarks to the Author):

The paper reports the production of a solid H₂-H₂O compound in the C0 phase and recovered the sample at room pressure and liquid nitrogen temperature. The H₂ rotational and OH and H₂ stretching vibrational spectra were used to monitor the presence of H₂. XRD was also used to probe the positions of the oxygen atoms in the structure. The Raman spectra and overall results are consistent with the proposed trigonal structure (space group P3112),The C0 structure appears to be stable even when emptied. The structure of the filled and empty C0 ice are essentially the same.

Several adsorption isotherms, deriving the amount of adsorbed H₂ as a function of pressure, at several different temperatures between 15 to 80 K are presented in Figure4. A large hysteresis and other kinetics effects are evident at the lower temperatures.

There is some interesting science in the paper. However, the isotherm evidence for a potential link to H₂ storage is quite weak. This potential application was quoted in the abstract 'Given these striking properties, we believe that ice XVII may be of large interest also for application in hydrogen storage, much more than what hydrogen clathrate hydrate have been in the past decade.' The isotherm data in Figure 4 are only given at low temperature (15-80K). Therefore the current evidence presented is weak. Ambient temperature isotherm measurements are required for justification of H₂ storage applications together with deliverable H₂ capacity.

A possible use of this material for hydrogen storage purposes obviously would require temperatures much lower than ambient, around liquid nitrogen temperatures, since the sample is mechanically stable only below 110-120 K. Anyhow, the importance of this material for hydrogen storage is not the main argument of this work, and has been further mitigated in this new version. The existence of a new form of ice, ice XVII, metastable at ambient pressure, that quickly adsorbs and releases gases seems to us a sufficient justification for the publication of this paper. A quantitative evaluation of the energy balance for application of ice XVII as hydrogen storage material will be attempted, but it is beyond the scope of this work.

Detailed answer to the Reviewers' comments.

Reviewer#1

We thank the Reviewer for this substantially positive comment. We have fulfilled his requirement of presenting the structure of ice XVII completing the Rietveld analysis of the neutron diffraction data we had obtained on OSIRIS, (RAL) on a empty deuterated sample. We have revised the manuscript essentially adding this information on the structure. The complete description and analysis of the neutron diffraction experiment, which was performed at different temperatures and involved several thermal treatment to study the effect of the guest, is beyond the scope of this paper and are the subject of a different work that we have submitted to J. Phys. Chem. Lett.. A reference to this has been added and a preprint copy of the submitted paper is included in this submission for completeness. This same preprint is going to appear soon on ArXiv.

The word metastable is used instead of stable everywhere throughout the text and in the title.

Reviewer#2

We thank the Reviewer#2 for this positive comment.